# Recycling of Nanocellulose from Polyester–Cotton Textile Waste for Modification of Film Composites

**DOI:** 10.3390/polym15153324

**Published:** 2023-08-07

**Authors:** Preeyanuch Srichola, Kuntawit Witthayolankowit, Prakit Sukyai, Chaiyaporn Sampoompuang, Keowpatch Lobyam, Prapakorn Kampakun, Raveewan Toomtong

**Affiliations:** 1Cellulose for Future Materials and Technologies Special Research Unit, Department of Biotechnology, Faculty of Agro-Industry, Kasetsart University, Chatuchak, Bangkok 10900, Thailand; 2Kasetsart Agricultural and Agro-Industrial Product Improvement Institute, Kasetsart University, Chatuchak, Bangkok 10900, Thailand; 3Faculty of Science, Department of Chemistry, Kasetsart University, Chatucak, Bangkok 10900, Thailand; 4Faculty of Agro-Industry, Department of Biotechnology, Kasetsart University, Chatuchak, Bangkok 10900, Thailand

**Keywords:** nanocellulose, textile waste, absorption, transmission, reflection, film composited

## Abstract

Textile waste has emerged as a critical global challenge, with improper disposal practices leading to adverse environmental consequences. In response to this pressing issue, there is growing interest in recycling textile waste containing cellulose as an alternative approach to reducing the impact of industrial waste on the environment. The objective of this research is to investigate the extraction and characterization of nanocellulose from polyester–cotton textile waste as a potential solution to address the growing concerns of waste management in the textile industry. To obtain nanocellulose, a comprehensive process involving alkaline sodium hydroxide (NaOH) treatment of the polyester–cotton textile (35% PET and 65% cotton) was employed, resulting in average yield percentages ranging from 62.14% to 71.21%. To achieve the complete hydrolysis of PET polyester in the blends, second hydrolysis was employed, and the optimized condition yield cotton fiber was 65.06 wt%, relatively close to the theoretical yield. Subsequently, the obtained cellulosic material underwent an acid hydrolysis process using 70 percent (*v*/*v*) sulfuric acid (H_2_SO_4_) solution at 45 °C for 90 min, resulting in nanocellulose. Centrifugation at 15,000 rpm for 15 min facilitated the separation of nanocellulose from the acid solution and yielded 56.26 wt% at optimized conditions. The characterization of the nanocellulose was carried out utilizing a comprehensive array of techniques, including absorption, transmission, and reflection spectra, and Fourier transform infrared. The characterization results provide valuable insights into the unique properties of nanocellulose extracted from textile waste. In this research, the obtained nanocellulose was mixed with PVA and silver nanoparticle to form biodegradable film composites as the reinforcement. In comparison, biodegradable film of PVA:nanocellulose 9.5:0.5 with silver nanoparticle 0.3 wt% and glycerol as a plasticizer exhibits better tensile strength (2.37 MPa) and elongation (214.26%) than the PVA film with normal cellulose. The prepared biodegradable film was homogeneous and had a smooth surface without the internal defect confirmed by the CT scan. This result opens avenues for enhancing the quantities of eco-friendly film composites, potentially replacing conventional plastic films in the future.

## 1. Introduction

In recent decades, textile consumption has been increasing to nearly double and the global fashion market is expected to rise from USD 1.5 trillion in 2020 to about USD 2.25 trillion in 2025 [1,2,3]. To accommodate the continuously rising demand, textiles manufacturing is also increasing and consequently influences climate, water and resources usage and the environment [4]. One of the major concerns is the accumulation of textile waste, and polyester–cotton blends have been a significant contributor to this environmental challenge [5,6]. Instead of assessing custom-made textiles and sustainability, the rapid growth of the textile industry, fueled by the global demand for clothing and apparel, has led to a significant surge in the consumption of polyester–cotton blends [7]. In general, a mixture of polyethylene terephthalate (PET) and cotton or natural fibers are the most common fabrics in the industry [8]. This human-made polyester fabric offers an affordable fabric with higher strength and durability while preserving cotton’s properties such as softness, breathability, and moisture-absorbent [9]. The massive consumption of polyester–cotton fabrics has given rise to ecological challenges, demanding urgent attention from stakeholders across industries and society. Due to the short life cycle of the polyester–cotton garments, 92 million tons of polyester–cotton textile waste was produced annually and contributed to the mounting burden on landfills and waste management systems [10]. Also, the surging demand for affordable blend fabric has led to labor exploitation and potential human health impacts due to the presence of harmful chemicals and dyes used during the processing. Thus, the current fashion industry has been acknowledged as unsustainable from both environmental and social perspectives [11]. To reduce the overall environmental implications of the value chain, the European Commission has developed the EU Strategy for Sustainable and Circular Textiles with conditions and incentives to achieve a circular ecosystem by 2030 [12]. This initiative promotes sustainability and efficiency throughout the textile value chain. In response to environmental concerns, recycling waste textiles through hydrolysis has been extensively studied [13,14].

To recycle polyester–cotton fabrics, the simplest method was mechanically defibrating without the addition of virgin fiber; however, this downcycling leads to inferior products [15]. Separating cellulose and PET from polyester–cotton blends presents significant challenges due to the complex nature of these materials. Polyester–cotton blends consist of both synthetic PET fibers in the polyester part, which is the minor component around 35 wt%, and natural cellulose fibers from cotton as a minor part. The distinct chemical compositions and physical properties of both PET fibers and cellulose contribute to the difficulties in finding a single solvent or method that can selectively dissolve one part while leaving another part intact and also make it difficult to achieve a clean and efficient separation process. Researchers have been exploring various techniques to address this challenge and develop efficient methods for separating cellulose and PET from polyester–cotton blends [16,17,18,19]. To recycle high-quality cotton from polyester–cotton fabrics, cotton and PET must be separated chemically through depolymerization using a different solvent, such as methylmorpholine *N*-oxide (NMMO), to dissolve cotton while removing PET fibers with filtration [20,21]. In this study, the opposite approach, maintaining cotton and depolymerizing the PET fibers, was used to obtain nanocellulose for making nanocomposite films. Since alkaline and acid hydrolysis has been shown the development of an efficient route for recycling cotton from mixed textile waste [22,23,24,25,26], we hereby report the optimized conditions to completely hydrolyze PET fiber from polyester–cotton fabrics while preserving cellulose and yield cellulose more than 93%. The obtained cellulose was processed into nanocellulose for nanocellulosic composite film. We also developed the biodegradable PVA silver nanocomposite film using nanocellulose obtained from polyester–cotton fabrics. This feasibility study showed the ability to synthesize silver nanomaterials as an eco-friendly, low-cost, and simple method. The developed PVA silver nanocomposite could have a wide range of applications in biomedical fields, such as antibacterial clothes and anti-infectious urinary catheters [27]. It can be integrated into antiseptic coverings for wounds and can be used in household applications such as textiles disinfection in water treatment and food storage [28]. Cellulose can be extracted and reduced for the production of nanocellulose. This can be done by many methods, such as mechanical, chemo-mechanical methods and enzyme techniques. Nanocellulose can be extracted from plant fibers or from algae. It is considered to be highly efficient due to its low thermal expansion and high modulus. It is biocompatible due to its low cytotoxicity, optical transparency and biodegradability. In a variety of areas, such as being used as a stabilizer, fat substitutes, emulsifiers, and food fibers, as well as used as a composite in the automotive industry [29,30,31,32].

Nowadays, the textile waste problem is a worldwide concern because some textile waste is not properly and methodically disposed of. The residue of these textile wastes causes a good deal of damage and impact on the environment. Therefore, the reuse of textile waste containing cellulose is another interesting alternative to reduce waste from such textiles [33,34]. Moreover, these celluloses are biodegradable. It does not cause residues in the environment and also promotes the utilization of waste from the textile industry as well. According to the Institute for Research and Development of Agricultural Products and Agro-industry has accumulated knowledge and focused on the production of biomass and biodegradable materials for use in paper and textiles for a long time [35,36,37]. Hence, the purpose of this study was to study the chemical separation of nanocellulose from textile waste and to study the properties of nanocellulose derived from textile waste by chemical methods, including studying the possibility of using nanocellulose derived from textile waste to be applied as a bio-based raw material in the production of film composited [38,39]. It is not only reducing the amount of textile waste but also reducing the process of limiting waste that can be harmful to the environment as well.

## 2. Materials and Methods

### 2.1. Material Preparation

Textile waste which was a blend of 35% PET and 65% cotton, was used in this study (Figure 1). In this experiment, the textile waste was treated with 0–20% alkaline sodium hydroxide (NaOH) at 85 °C for 2–4 h, and then the obtained textile waste with cellulosic material was hydrolyzed with 60% concentration of sulfuric acid (H_2_SO_4_) at 45 °C for 1.30 h. The obtained cellulosic material, 35 g, was loaded into the tube and was centrifuged (Refrigerated Centrifuge, 5430R, Eppendorf, Germany) with a speed of 15,000 rpm for 15 min to separate the nanocellulose from the acid solution in 56.26% yield. From additional research, it was found that in some cases, Benzyltributylammonium chloride (BTBAC) was used in accelerating the hydrolysis reaction [8,32]. However, the price of such substances is quite high. Then, the authors repeated the NaOH alkali hydrolysis experiment in the second step to dissolve the fiber. The obtained nanocellulose was dialyzed by using cellophane dialysis tubing at 12–14 kdalton, and then it was washed with distilled water until pH was equal to 7 (Figure 2). The suspended nanocellulose was then placed in the ultrasonic bath (Elma, Singen, Germany) before testing its properties.

### 2.2. Preparation of PVA Film Incorporated with Nanocellulose

The PVA (Mw = 89,000–98,000) concentration of 10 g/100 mL with a density of 1.19 g/cm^3^ was dissolved in deionized water and heated by using a hot plate and magnetic stirrer at 90–100 °C for 40 min. After obtaining a homogeneous solution, nanocellulose (*w*/*w* of PVA powder), AgNPs and glycerol were put into the homogeneous solution. The nanocomposite film was prepared by using a digital overhead mixer (IKW rw 20). After mixing for 30 min, the resultant hydrogel mixture was gently poured into a Petri dish with an average diameter of 14 cm; and allowed to settle down slightly for 60 min at room temperature to form a film. The film was finally dried in an oven at 45 °C for 12 h (Figure 3).

### 2.3. Characterizations

The standard methods were studied and modified to evaluate the PVA film incorporated with nanocellulose properties produced in a laboratory.

#### 2.3.1. Photophysical Properties

Ultraviolet-Visible Absorption, Transmission and Reflection Spectra were Recorded on Absorption Spectrometer with a Measurement Range of 325–800 nm (V-730 spectrophotometer, JASCO, Tokyo, Japan). The transmission was measured with the measurement mode of the transmission spectra; interval scanning data: 1.0 nm, scanning rate 100 nm/min, bandwidth: 2.0 nm, measurement range 325–800 nm. The measurement mode of the reflection spectra; interval scanning data: 1.0 nm, scanning rate 100 nm/min, bandwidth: 5.0 nm, measurement range 325–800 nm, respectively. The materials were placed in a quartz cell and set to record the spectra.

#### 2.3.2. Infrared Radiation Absorption Property

Fourier Transform Infrared (Nicolet IR200 FT-IR Infrared Spectrometer, Thermo Scientific, Waltham, MA, USA) was used in the study for checking the infrared radiation absorbance with measurement mode of the absorption spectra; interval scanning data: 0.02 of intensity, measurement range 450–4000 cm^−1^. The materials were placed on a cell and set to record the spectra.
$$A=\log\frac{I_{0}}{I}=\varepsilon \mathrm{bc}$$
where
A=Absorbanceε=Molar absorptivity unit dm^3^ cm^−1^ g^−1^c=Concentration unit g dm^−3^ or mol L^−1^ or molarI_0_=Light intensityI=Light intensity when passing the sampleb=Thickness of the sample

#### 2.3.3. Basis Weight

Basis Weight of PVA Film Incorporated with Nanocellulose was Tested According to ISO 536 (Determination of Grammage).
$$G=\frac{m}{A}$$
where
G=Basis weight (g/m^2^)m=Mass (g)A=Area (m^2^)

#### 2.3.4. X-ray Computed Tomography

X-ray Computed Tomography Scanning and Composite Materials with nano and micro-focus CT scanning (Phoenix V|tome|x M240, Waygate, Hurth, Germany), the data will be captured at incredibly high resolution. CT scanning is an extremely valuable tool in materials research. 3D image data was acquired using a Waygate Phoenix V|tome|X-ray computed tomography scanner. An accelerating voltage of 50 kV, a current of 230 µA, and a power of 11.5 W were set. The voxel size was 0.6 µm, and an exposure time of 2 s with 1000 projections was used, resulting in a total scan time of 150 min. The raw data was reconstructed using Phoenix Datos|x software. The reconstructed 3D image data was further processed and analyzed using VG Studio Max software.

#### 2.3.5. Mechanical Properties

Mechanical Properties Tensile Strength and Percentage Elongation at Break of r-PET filaments were measured following the standard method of ASTM D3822 using a tensile tester (AGS5kN, Shimadzu, Tokyo, Japan). The prepared PVA films incorporated with nanocellulose were cut to 1.5 × 11 cm and gripped to the tensile tester. The loadcell in this study was 50 N, and the pulling speed was 50 mm/s. Each mechanical property test was repeated four times.

## 3. Results

The researchers planned to separate the composition of textile waste, which contained polyethylene terephthalate (PET) and natural fibers. The primary hydrolysis was performed using NaOH at different concentrations. The collected samples after the alkaline hydrolysis reaction were filtrated with No. 1 filter paper, and then the collected water measured the acidity–alkalinity after the reaction finished. The percent yield of cellulose extracted are depicted in Table 1.

The primary hydrolysis using alkaline NaOH at different concentrations shows that the alkaline hydrolysis of the textile waste gave a PET mixed cellulose yield of 60–72 wt% after 2 h and 4 h, with 0–20% NaOH in the solution (A0-20 and B0-20). From the observation with the naked eye in conjunction with the measurement of the pH base using litmus paper, it can be found that A0 and B0 showed pH equal to 7, which can be preliminarily concluded that only thermal hydrolysis has a very low reaction rate. However, when increasing the NaOH concentrations, a significant difference in the PET mixed cellulose yield was found in the experiment (from 62.14 to 69.88 wt%). This is because of the incomplete hydrolysis of PET from textile waste. However, at 20%, NaOH revealed the highest yield at 71.76 wt%. Hence, A20 and B20 were selected to be the raw materials in the second step of NaOH hydrolysis. The second step showed complete hydrolysis of PET from the textile waste as the cellulose yield at 65.06 wt% for A20/20 and 61.75 wt% for B20/20. Due to the second hydrolysis in each condition, the second hydrolysis of A20/20 was the optimized time and yielded cellulose slightly higher than B20/20. It is noted that at condition B20/20, the yield of obtained cellulose is lower than the theory yield, which is 65 wt%, and it can be explained that the hydrolysis of polyester–cotton fabric occurring twice at 4 h (8 h in total) was too harsh and some of the cellulose degradation occurred. Both of the obtained cellulose were white in color, and the fiber of cellulose was still intact. The obtained cellulose fiber was processed to nanocellulose via acidic hydrolysis followed by centrifugation and sonication (Figure 2).

The chemical mechanism of hydrolysis of 35% synthetic and 65% natural fibers with NaOH into diasodium terephthalate salt and ethylene glycol is shown in Figure 4. The primary and secondary hydrolysis of PET from the textile waste as the cellulose was showed in Figure 5 and Figure 6. Due to the ester moiety in the PET fiber, it is prone to basic hydrolysis and can be cleaved with sodium hydroxide at high temperatures to yield carboxylate salt and diol. After the reaction is acidified, sodium terephthalate will be protonated and yield valuable monomer terephthalic acid [40,41].

To evaluate the obtained nanocellulose from secondary hydrolysis for both A20/20 and B20/20, the characteristic functional group of nanocellulose must be illustrated. The differences in function groups among various absorbents were elucidated through Fourier-transform infrared (FTIR). The analysis revealed distinct peaks corresponding to specific molecular interactions and binding phenomena. The most prominent peak observed at 3434.75 cm^−1^ was attributed to the alicyclic ether with a strong O-H stretching vibration. Furthermore, stretching vibrations of CH, CH_2_, and CH_3_ bonds by hydroxyl groups were identified within the range of 2356.26 to 2923.18 cm^−1^. Peaks in the 1667–2000 cm^−1^ region indicated the presence of aromatic hydrocarbons. Additionally, symmetric stretching of the C=C group was observed at 1635.21 cm^−1^, while asymmetric stretching of the C-O group was evident at 1113.12 cm−1, both of which signify the presence of the alicyclic ether (Figure 7 and Figure 8). Both nanocellulose obtained from the second hydrolysis, A20/20 and B20/20, showed quite similar signature peaks, resembling that despite the harsh hydrolysis at a total of 8 h (B20/20), the nanocellulose was still intact, and the functional groups did not get modified. Comparative analysis of the FTIR spectra demonstrated that the absorption intensities of each functional group were contingent on the amount of substance present in the nanocellulose sample. As each sample differed in size, variations in the spectrum emerged with distinct absorption values. This phenomenon can be attributed to the quantized nature of infrared absorption, wherein the absorbed frequency must correspond to the bond vibration frequency to facilitate infrared radiation absorption.

In this study, a comprehensive analysis of the nanocellulose’s optical properties was conducted through the assessment of absorption, transmission, and reflection spectra. The resulting nanocellulose exhibited a prominent absorption peak at approximately 400 nm in the absorption spectrum, albeit with a slight reduction observed across all samples. The absorption behavior of nanocellulose obtained from A20/20 and B20/20 demonstrated similar trends, potentially indicative of the luminescence intensity within each specimen. It is important to note that due to limitations in the laboratory equipment, the authors could not ascertain whether nanocellulose would exhibit absorption at higher energy (UV range). Moreover, the transmission and reflection spectra also showcased a peak at 400 nm, consistent with the absorption spectrum (Figure 9). Additionally, the nanocellulose extracted from textile waste exhibited luminescence in a captivating blue hue when exposed to UV light (Figure 10). This investigation into the optical characteristics of nanocellulose holds significant implications for its potential applications in various fields, such as optoelectronics and photonics. Further studies could be pursued to gain deeper insights into the luminescence properties and their correlation with the nanocellulose’s structure and composition. The ability of nanocellulose to absorb and emit light at specific wavelengths opens up avenues for the development of advanced materials with tailored optical properties.

In this study, casting technology was employed to produce biodegradable films by blending homogeneous substances in a water solvent. The process involved the direct preparation of the prepared biodegradable film of Polyvinylalcohol (PVA) filled with AgNPs (silver nanoparticles) through the alternate deposition of mixed dispersions of nanocellulose, AgNPs, and glycerol. The resulting film exhibited a smooth surface and demonstrated a classical linear increase in film thickness with each assembly step. Additionally, the composite films showcased the successful recycling of nanocellulose derived from polyester–cotton fabrics for the modification of the PVA film. Following 24 h of drying in the oven, the PVA filled with AgNPs, glycerol, and nanocellulose exhibited thickness and basis weight ranging between 0.206–0.291 mm and 397.22–406.25 g/m^2^, respectively.

The convergence of nanocellulose and AgNPs within the PVA matrix holds the potential to augment the film’s properties, introducing desirable characteristics such as enhanced mechanical strength and antibacterial attributes. This biodegradable film presents a promising avenue for the utilization of nanocellulose extracted from textile waste, contributing to the development of sustainable materials with diverse applications in biomedical, packaging, and other industrial domains. For the nanocellulose, the robust interfibril interactions within the cellulose structure contribute to the high transparency property and strong tensile strength. However, the disadvantage of nanocellulose-based films is the low elongation at the point of fracture which limits the applications that necessitate a balance between both flexibility and strength. By leveraging the versatile casting technology and synergistic combination of nanocellulose, AgNPs, and PVA, we can further explore the optimization of film properties to meet specific requirements for different applications. As shown in Table 2, the biodegradable film prepared from PVA and normal cellulose (ratio 95:5) with AgNPs 0.3 wt% exhibited poor tensile strength (1.71 MPa) and elongation property (43.90%). It is suggested that normal cellulose did not blend well with PVA in the casting step resulting in uneven film and lower physical properties. To strengthen the film, the biodegradable film was cast with nanocellulose in the same manner and ratio and the physical properties were significantly improved. The obtained biodegradable film with nanocellulose has a tensile strength of 2.37 MPa and elongation of 214.26%. It is due to the good dispersion of nanocellulose during the casting process resulting in homogeneous film without cracking on the surface of the film.

After X-ray computed tomography scanning and composite materials with nano and micro-focus CT scanning, it was found that the PVA filled with AgNPs, glycerol, and nanocellulose exhibited sandwich layers according to the density, as shown in Figure 11B. It was noted that the specimen was stained in the absence of any cracking. The development and distribution of fibers from these sandwich layers showed that the two fibers, apart from the fact that these splits, are closely correlated with the high density (1.60 g/cm^3^) of the nanocellulose inside the film composited sample. In the case of PVA, the authors assume that it is presented not only in the middle but also on the top and bottom of the layer, with its density at 1.19 g/cm^3^. Figure 12A reveals a longitudinal section of the film composited. It displayed the film composited smoothness which indicated that the film composited was successfully made in a laboratory. The cross-section was scanned, as shown in Figure 12B. The film composited clearly seen sandwich layers according to the density, as the author said above.

To investigate the defect of cracking inside the biodegradable film with nanocellulose, X-ray computed tomography scanning and composite materials with nano and micro-focus, CT scanning was employed. The PVA filled with AgNPs and nanocellulose revealed the presence of distinctive sandwich layers delineated by varying density, as depicted in Figure 11B. Notably, the film displayed an absence of any cracking, indicating structural integrity. Further examination of the sandwich layers elucidated the development and distribution of fibers within the nanocomposite film. It was observed that two types of fibers were closely correlated with the high density (1.60 g/cm^3^) of the nanocellulose integrated into the film composite sample. Conversely, the PVA component was assumed to be present not only in the middle layer but also on the top and bottom, contributing to an overall density of 1.19 g/cm^3^. A longitudinal section of the film composite (Figure 12B) offered insight into the smoothness and cohesion achieved in the laboratory-made nanocomposite film. Furthermore, a cross-sectional scan (Figure 12B) confirmed the presence of sandwich layers in accordance with the density-related findings mentioned earlier. These CT scanning results provide valuable visualization of the internal structure and morphology of the PVA filled with AgNPs, and nanocellulose nanocomposite. The imaging reveals the interplay of nanocellulose and PVA within the film, along with the distribution of AgNPs. In-depth characterization serves as a foundation for understanding the nanocomposite’s material behavior and mechanical properties. Thus, the prepared biodegradable PVA and nanocellulose film holds the promise of contributing to the advancement of eco-friendly materials and the transition towards a more sustainable materials for diverse industrial applications.

## 4. Conclusions

The successful preparation of nanocellulose from polyester–cotton waste was achieved through a two-step process involving alkaline sodium hydroxide (NaOH) treatment and acid hydrolysis. The study resulted in a significant cellulose yield of 65.06 wt%, which is almost a theoretical yield, and resembled the complete hydrolysis of PET from the textile waste. The obtained nanocellulose revealed intriguing optical properties, evident from the absorption, transmission, and reflection spectra, which displayed a prominent peak around 400 nm. Moreover, under UV light, all nanocellulose samples exhibited a distinct blue coloration. The Fourier-transform infrared spectroscopy results further confirmed the successful hydrolysis of PET content in the polyester–cotton waste and yield cellulose fiber, establishing the effectiveness of the recycling process. The biodegradable film prepared from PVA and normal cellulose (95:5 ratio) with 0.3 wt% AgNPs exhibited poor tensile strength (1.71 MPa) and elongation (43.90%). However, when the film was reinforced with nanocellulose in the same ratio, the physical properties were significantly improved, resulting in a higher tensile strength (2.37 MPa) and elongation (214.26%). The well-dispersed nanocellulose during the casting process contributed to a homogeneous film without surface cracking, and the thickness and basis weight were 0.206–0.291 mm and 397.22–406.25 g/m^2^, respectively. By employing X-ray computed tomography scanning and composite materials, the nanocomposite films exhibited intriguing sandwich-like layers, as evidenced by the varying densities of the components. The successful recycling of nanocellulose from textile waste represents a significant advancement, as it can be utilized as an eco-friendly material in conjunction with polyvinyl alcohol to fabricate film composites. These biodegradable film composites have the potential to serve as viable alternatives to plastic films in the future.

## Figures and Tables

**Figure 1 polymers-15-03324-f001:**
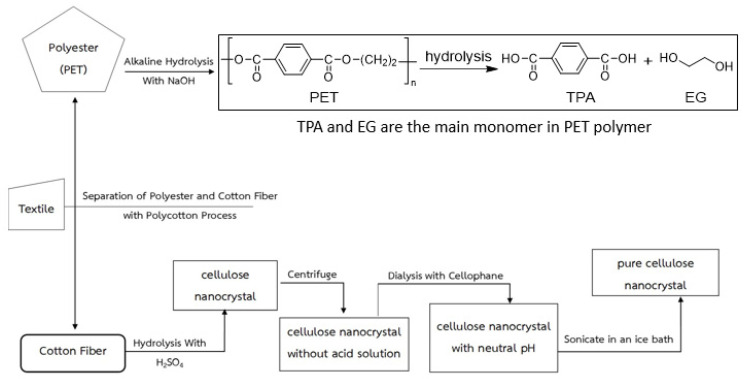
Process of polyester–cotton textile separation.

**Figure 2 polymers-15-03324-f002:**
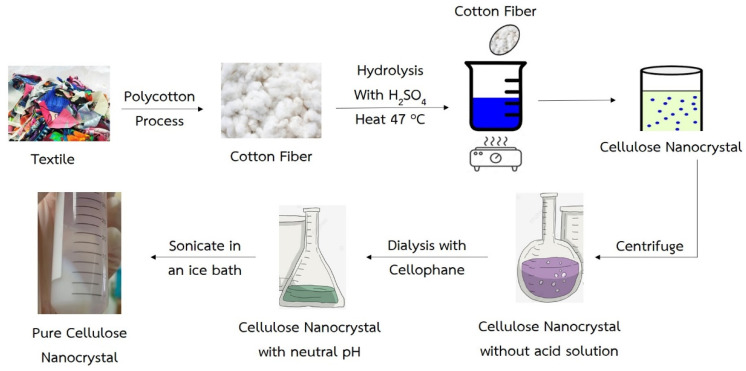
Nanocellulose preparation from polyester–cotton via the two steps of hydrolysis followed by centrifugation and sonication.

**Figure 3 polymers-15-03324-f003:**
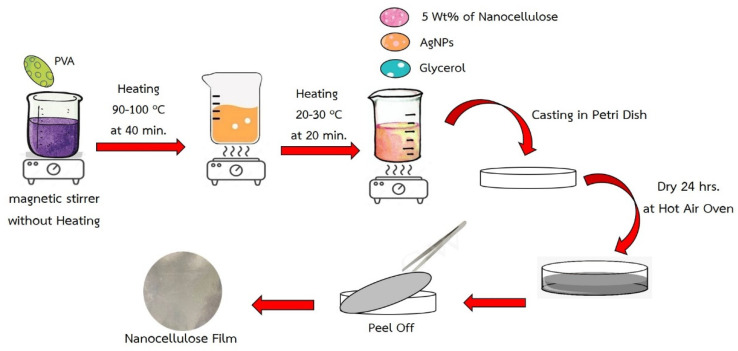
Preparation of PVA film incorporated with nanocellulose.

**Figure 4 polymers-15-03324-f004:**
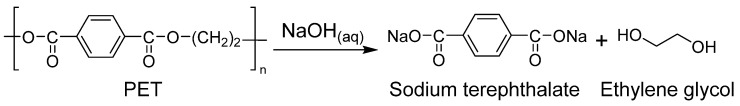
Chemical mechanism of hydrolysis of 35% synthetic and 65% natural fibers with NaOH into diasodium terephthalate salt and ethylene glycol.

**Figure 5 polymers-15-03324-f005:**
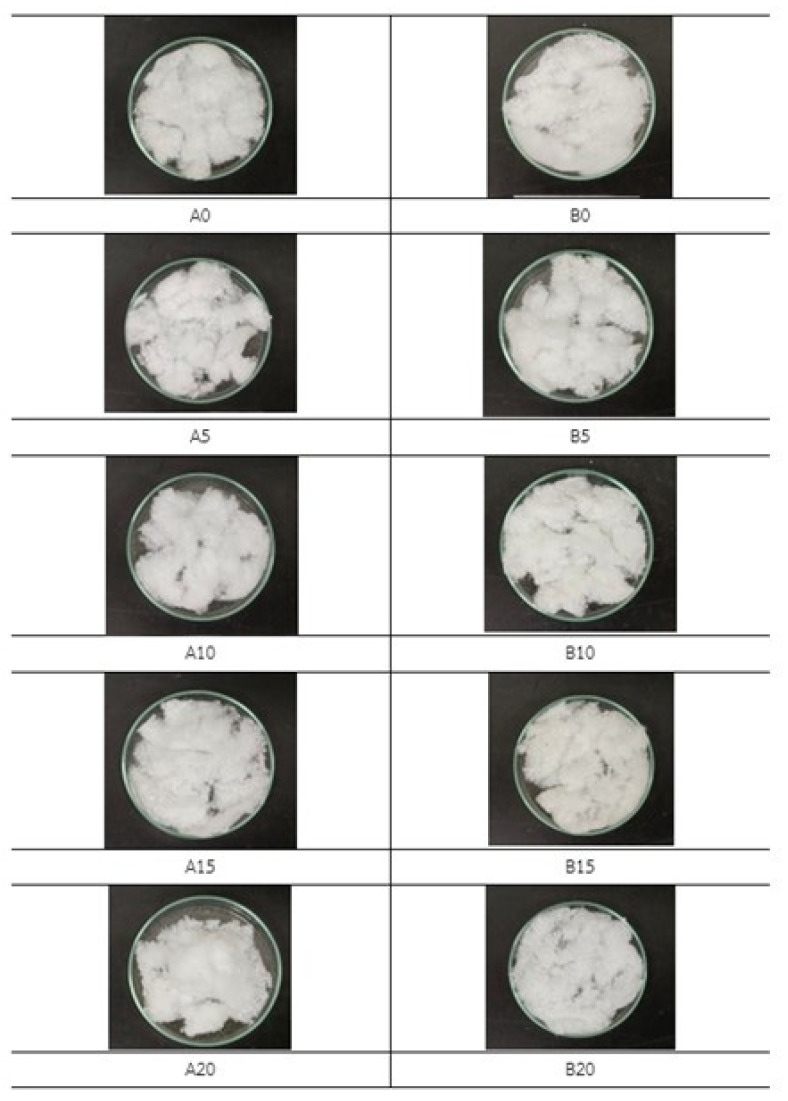
The primary hydrolysis using alkaline NaOH 0–20 wt% with condition A for 2 h and condition B for 4 h.

**Figure 6 polymers-15-03324-f006:**
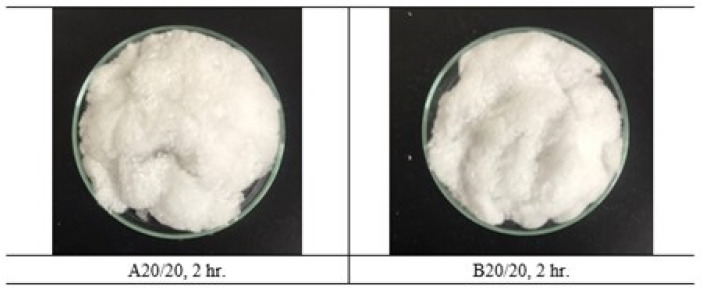
Secondary hydrolysis of each A20 and B20 using alkaline NaOH at 20% concentration. The complete hydrolysis of A20/20 and B20/20 yield cellulose 65.07 and 61.75 wt%, respectively.

**Figure 7 polymers-15-03324-f007:**
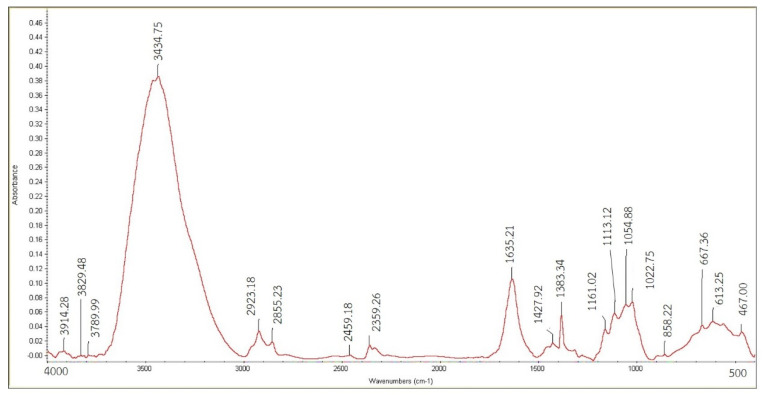
FTIR spectroscopy of cellulose from condition A20/20.

**Figure 8 polymers-15-03324-f008:**
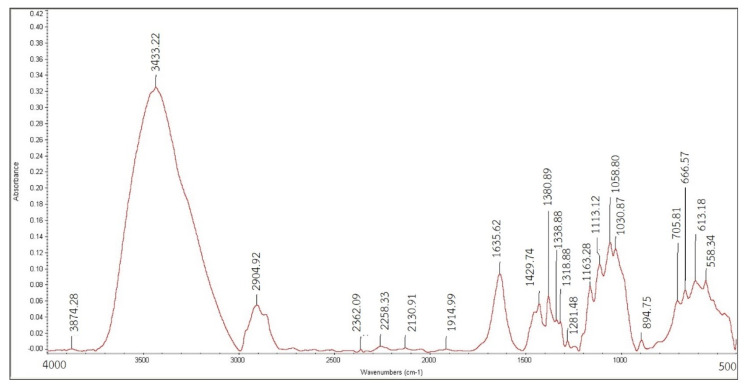
FTIR spectroscopy of cellulose from condition B20/20.

**Figure 9 polymers-15-03324-f009:**
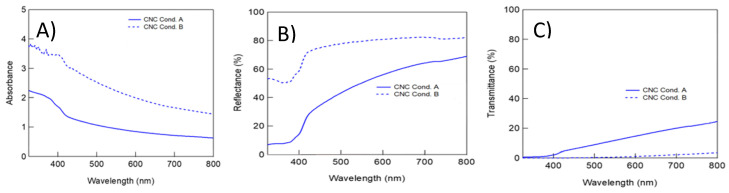
Photophysical properties of nanocellulose extracted from condition A20/20 (bold line) and B20/20 (dash line). (**A**) Absorption, (**B**) transmittance, and (**C**) reflectance.

**Figure 10 polymers-15-03324-f010:**
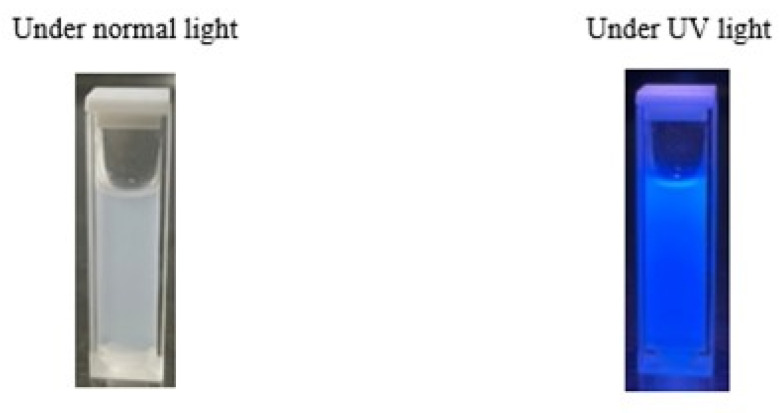
Nanocellulose extracted shows a blue color under UV light.

**Figure 11 polymers-15-03324-f011:**
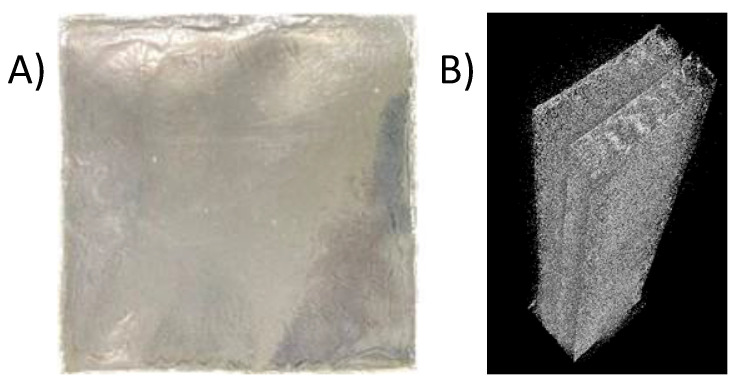
The prepared biodegradable film. (**A**) PVA and nanocellulose film filled with silver nanoparticles nanocomposite. (**B**) Image spectrometer of film (**A**) by X-ray computed tomography scanning.

**Figure 12 polymers-15-03324-f012:**
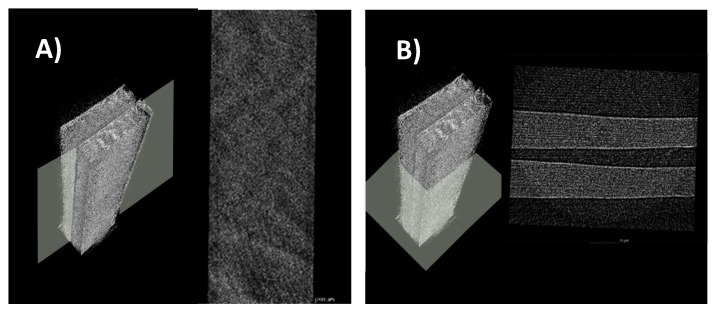
CT spectrometer of PVA and nanocellulose film. (**A**) Longitudinal section of the film composite. (**B**) Cross-section of the film composite.

**Table 1 polymers-15-03324-t001:** Percentage yield of the primary hydrolysis using alkaline sodium hydroxide (NaOH) at different concentrations. A and B represent the hydrolysis of 2 and 4 h, respectively. The number after A and B represent the concentration of NaOH (wt%) for each hydrolysis.

No. of Samples	NaOH (%)	Temp. (°C)	Time (h)	PET Mixed Cellulose Yield (%)
A0	0	85	2	62.14
A5	5	85	2	69.88
A10	10	85	2	71.21
A15	15	85	2	69.36
A20	20	85	2	71.76
B0	0	85	4	67.76
B5	5	85	4	68.73
B10	10	85	4	69.21
B15	15	85	4	68.32
B20	20	85	4	70.83

**Table 2 polymers-15-03324-t002:** The average tensile and elongation of PVA films incorporated with normal cellulose and nanocellulose. Each test was repeated four times and reported with ± SD.

Samples	Average Tensile (MPa)	Average Elogation (%)
PVA film incorporated with 5 wt% cellulose and 0.3 wt% AgNPs	1.71 ± 0.38	43.90 ± 20.20
PVA film incorporated with 5 wt% nanocellulose and 0.3 wt% AgNPs	2.37 ± 0.32	214.26 ± 44.91

## Data Availability

Data are contained within the article.

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
