# Peer review of "Recycling of Nanocellulose from Polyester–Cotton Textile Waste for Modification of Film Composites"

_polymers, 2023, doi:10.3390/polym15153324_

Round 1
Reviewer 1 Report
Dear authors,
Overall, this paper is interesting. However, in my opinion, it needs some significant adjustments. To improve its value, I include detailed comments below.
I think the introduction is written quite well. The justification for undertaking the research and the purpose of the work are also correct. I think that you should also indicate the possibilities of using cellulose, which can be used to produce various biocomposites. For example, microcrystalline cellulose is added to biocomposites made from various fruit pomace or waste: See the following articles: "A comprehensive review on cellulose, chitin, and starch as fillers in natural rubber biocomposites", "Properties of Biocomposites from Rapeseed Meal, Fruit Pomace and Microcrystalline Cellulose Made by Press Pressing: Mechanical and Physicochemical Characteristics”. This will better underline the legitimacy of your research.
Line 99: Enter the name of the centrifuge and the overload occurring during centrifugation, e.g. 24 x g. Moreover, each apparatus used in the research should have a description (name: manufacturer, city, country). This should be taken into account throughout the methodology.
Figure 1, 2, 3: I think the font on the pictures should be a little bigger. Especially above the arrows.
Table 2 and 3: However, the caption "Figure" is better here than the table. Then, an explanation or brief description of all samples/acronyms should be provided below the figure.
Figures 5 and 6 are also hard to read. The font should be increased.
Figure 9, 10. Use arrows/circle…etc to mark the significant places used for the analysis/description of this image.
Line 251: Are the results close to or far from expected? Compare the results with studies by other authors. In my opinion, the discussion of the results is poor
Line 257: Add one more forward looking one. A more prospective proposal.
Author Response
Respond to reviewer 1
Overall, this paper is interesting. However, in my opinion, it needs some significant adjustments. To improve its value, I include detailed comments below.
I think the introduction is written quite well. The justification for undertaking the research and the purpose of the work are also correct. I think that you should also indicate the possibilities of using cellulose, which can be used to produce various biocomposites. For example, microcrystalline cellulose is added to biocomposites made from various fruit pomace or waste: See the following articles: "A comprehensive review on cellulose, chitin, and starch as fillers in natural rubber biocomposites", "Properties of Biocomposites from Rapeseed Meal, Fruit Pomace and Microcrystalline Cellulose Made by Press Pressing: Mechanical and Physicochemical Characteristics”. This will better underline the legitimacy of your research.
Thank you for the compliments for introduction and for your suggestion for improvement. These 2 references was added to the revised introduction part. We have added more introduction about polyester-cotton recovery and, also, biodegradable film. We hope this will highlight our research results.
Line 99: Enter the name of the centrifuge and the overload occurring during centrifugation, e.g. 24 x g. Moreover, each apparatus used in the research should have a description (name: manufacturer, city, country). This should be taken into account throughout the methodology.
Thank you for the point, we added the description of each apparatus. The overload in centrifuge tube was added as per suggestion.
Figure 1, 2, 3: I think the font on the pictures should be a little bigger. Especially above the arrows.
Thank you for the comment, these figures were fixed as suggestion.
Table 2 and 3: However, the caption "Figure" is better here than the table. Then, an explanation or brief description of all samples/acronyms should be provided below the figure.
Thank you for the comment, we change the table to figure as suggestion.
Figures 5 and 6 are also hard to read. The font should be increased.
Thank you for the comment, these figures were fixed as suggestion.
Figure 9, 10. Use arrows/circle…etc to mark the significant places used for the analysis/description of this image.
Thank you for the comment, these figures were fixed as suggestion.
Line 251: Are the results close to or far from expected? Compare the results with studies by other authors. In my opinion, the discussion of the results is poor.
Thank you for the comment, we have added more results to improve the impact of this paper, and we hope they meet your standard. In this research, we aim to recycle the polyester-cotton ester to obtain the nanocellulose and utilized it to strengthen biodegradable film. Compare to normal cellulose, the films with nanocellulose expected to exhibit better physical properties, which are tensile strength and elongation. Also, we encountered the difficulty for the film preparation and we optimized the conditions to get the thinnest film possible but also strong enough for actual applications.
Line 257: Add one more forward looking one. A more prospective proposal.
Thank you for the comment, we have added more results on physical properties of the prepared bridgeable films. The nanocellulose in the film gave the better tensile strength and elongation of the films compared to normal cellulose. We have successfully recovered and prepared nanocellulose from the polyester-cotton waste and turned it to biodegradable films. Furthermore, the prepared biodegradable films have the anti-microbial activities due to the silver nanoparticle. In contrast, the film without silver nanoparticle got black dots of fungi when they were leaved in the storage for 10 days, due to the moisture and temperature of our country. This anti-microbial application of the films will be studied in the future.

Reviewer 2 Report
The experimental article "Recycling of nanocellulose from TC textile waste for modification of film composites polyvinyl alcohol" is devoted to the recycling of cellulose from textile waste for the subsequent production of nanocellulose, which, in turn, is used to make film composites polyvinyl alcohol. In many ways (subject, research process), this short article corresponds to the Polymers publication. The relevance of the topic of recycling textile (cotton) waste into nanocellulose is beyond doubt. But the article should be completely revised by the authors with the elimination of a large number of shortcomings and errors. It is absolutely obvious that these shortcomings are connected either with the haste in the process of preparing this material, or with the need to shift the purpose and objectives of the study, which is typical when changing the place of publication.
Notes:
1. Delete "TC" from the title of the article, since this abbreviation is missing.
2. Rewrite the introduction, paying attention to the difficulty of isolating cotton pulp from textile waste, which is a mixture of polyethylene terephthalate (PET) and cotton or natural fibers.
3. Rewrite the introduction, eliminating the lyrical reasoning about modern fashion.
4. Find a reason to cite publications of 2023 that are similar in content to this experimental work, in particular, for example, to note new areas of use of textile waste.
5. Find a replacement out of respect for the Polymers edition of the term "polycotton" coined by the authors.
6. Answer the question: why did sodium hydroxide treatment suddenly become “the biochemical separation of nanocellulose from textile waste”? And if there is no explanation for this, then correct it everywhere in the text!
7. Rewrite the abstract to clearly indicate the subject of the study, the processes carried out with it, and the novelty of the results obtained by the authors.
8. Please note that tables 2 and 3 (they contain photos), as well as Figure 9 (two photos) do not have a semantic load. It is very difficult to trace the changes.
9. Check link 13.
Author Response
Respond to reviewer 2
The experimental article "Recycling of nanocellulose from TC textile waste for modification of film composites polyvinyl alcohol" is devoted to the recycling of cellulose from textile waste for the subsequent production of nanocellulose, which, in turn, is used to make film composites polyvinyl alcohol. In many ways (subject, research process), this short article corresponds to the Polymers publication. The relevance of the topic of recycling textile (cotton) waste into nanocellulose is beyond doubt. But the article should be completely revised by the authors with the elimination of a large number of shortcomings and errors. It is absolutely obvious that these shortcomings are connected either with the haste in the process of preparing this material, or with the need to shift the purpose and objectives of the study, which is typical when changing the place of publication.
Thank you for your understanding to our situation, we are really appreciated it. We hope that this revision will meet your standard.
Notes:
- Delete "TC" from the title of the article, since this abbreviation is missing.
Thank you for the comment, we have replaced "TC" with polyester-cotton.
- Rewrite the introduction, paying attention to the difficulty of isolating cotton pulp from textile waste, which is a mixture of polyethylene terephthalate (PET) and cotton or natural fibers.
Thank you for the comment, we have added more introduction regarding the isolation products of polyester-cotton.
- Rewrite the introduction, eliminating the lyrical reasoning about modern fashion.
Thank you for the comment, we have fixed the introduction part.
- Find a reason to cite publications of 2023 that are similar in content to this experimental work, in particular, for example, to note new areas of use of textile waste.
Thank you for the suggestion, we have updated recent publications in references. We really hope that they have meet your standard.
- Find a replacement out of respect for the Polymers edition of the term "polycotton" coined by the authors.
Thank you for the comment, we have fixed the manuscript as per suggestion.
- Answer the question: why did sodium hydroxide treatment suddenly become “the biochemical separation of nanocellulose from textile waste”? And if there is no explanation for this, then correct it everywhere in the text!
Thank you for the comment, we found that this is the mistake and corrected it following comment.
- Rewrite the abstract to clearly indicate the subject of the study, the processes carried out with it, and the novelty of the results obtained by the authors.
Thank you for the comment, we have rewrite the abstract and highlight our finding and process. We have successfully extracted and prepared nanocellulose from polyester-cotton waste and the prepared nanocellulose was used as strengthened for the biodegradable film. Compare to normal cellulose, the films with nanocellulose expected to exhibit better physical properties, which are tensile strength and elongation. The conditions were optimized to get the thinnest film possible but also strong enough for actual applications. Even though our finding is not the breakthrough results, we really hope that they can contribute to the field of material science.
- Please note that tables 2 and 3 (they contain photos), as well as Figure 9 (two photos) do not have a semantic load. It is very difficult to trace the changes.
Thank you for the comment, we fixed the description of these figures to add more information.
- Check link 13.
The references were fixed. We decided to remove this reference and add more recent one.

Round 2
Reviewer 2 Report
The experimental article "Recycling of nanocellulose from polyester-cotton textile waste for modification of film composites polyvinyl alcohol" has undergone very large changes. The authors patiently eliminated all the comments of the reviewer, including changing the title, completely rewriting the abstract and conclusion, and redoing the figures. The list of used literature has also undergone major changes.
I believe that the article should be carefully proofread by the authors after the removal of redundant text and can be published.